# Probability of Winning the Tender When Proposing Using BIM Strategy: A Case Study in Saudi Arabia

Mahmoud Majzoub [1] and Ahmed Eweda [2,*]

1   Arabian Towers Projects Contracting Co., Dammam 32210, Saudi Arabia; majzoob555@gmail.com
2   Corporate Asset Management, London, ON E1 6AN, Canada
*   Correspondence: aeweda@london.ca

**Abstract:** The procurement process is one of the most important phases in any project life cycle, particularly when it comes to selecting the right contractor for the job. Awarding the contract to the best bid proposal is a critical step to ensure the greatest value. BIM has been recognized as not only a geometric modelling of buildings, but also, it facilitates the different stages in management of construction projects. The purpose of this paper is to study the impact of using Building Information Modeling (BIM) in the tendering process from the contractor's perspective, based on a probability model able to predict winning probability, regardless of relative weight. The main objective of this research is to measure the likelihood of winning a tender in the case of implementing BIM strategy, compared with contractors who do not use BIM. The research uses a literature review, surveys, and interviews with experts to develop a model that predicts the probability of winning a contract; this is determined by measuring the BIM impact on each selection criterion in a multicriteria selection process using the Analytical Hierarchy Process (AHP) to develop a probability-based model. The results of the survey and the interview show that BIM strategy has a variant influence on the score the contractor could have on each of them raising the probability of winning the tender. The main result of this paper is the property-based model, which is able to predict BIM winning probability regardless of relative weight, which can be applied in any country. Nonetheless, the Saudi case study shows that utilizing BIM when proposing could increase the winning probability by up to 9.42% in the case of Quality-Based Selection (QBS), and to 5.5% in the case of Cost-Based Selection (CBS).

**Keywords:** BIM strategy; BIM technology; tender selection criteria; project delivery methods; probability-based model; probability of winning the tender



## 1. Introduction

One of the most important phases during construction project management processes is the tendering process. During the tendering process, especially in mega structures, there could be a high probability that bidders could go for innovative strategies. There is no doubt that owners are keen to use these strategies for their own good, and equally, contractors are in a continuous race to provide the best tactics to use their accumulative experience to establish a competitive advantage; nonetheless, BIM strategy could be a path for both parties to insure the maximum profit possible [1,2].

Hence, to win as many project contracts as possible, a contractor should provide a favorable offer to the owner, represented through quality, cost, and duration [3]. Moreover, there is an international trend to use Building Information Modeling (BIM) as a tool in design, construction, and even facilities management project stages. This tool offers architects and engineers the ability to avoid conflicts between design disciplines, while modeling and constructing buildings.

Nevertheless, while BIM may be an effective tool to model the project and establish a strong base for the design process, there should be an efficient mechanism that would effectively use the BIM properties in the contractor's favor [4], leading to the importance of

identifying BIM and BIM strategy to differentiate between these two terms. While BIM is an infinite dimensional modeling process that enables project participants and professionals to design, construct, and operate the building with data-driven insights, BIM Strategy is a methodology of BIM implementation through selection and investing in processes, technology, tools and people to fulfill the set goal [5].

The objective of this paper is to study and determine the impact of using BIM strategy on the probability of winning the tender. This objective will be accomplished by studying the characteristics of BIM and its strategy from international and local perspectives, identifying Tender Selection Criteria (TSC), determining the contribution of BIM in tender selection criteria, and developing a model that measures the probability of winning the contract when proposing using BIM strategy. This paper will assume that the tender is applied in cases of Quality-Based Selection (QBS) or Cost-Based Selection (CBS), which have been applied in both sectors, private and public; hence, bidding through BIM is more likely to have a higher cost compared to contractors who do not use it—BIM implementation requires extra overhead costs for experts and technologies [6–8]. Saudi Arabia will be the spatial limitation of this study; it is one of the fastest developing countries with a tremendous amount of construction projects, and is in the phase of creating its own regulation in BIM use, where projects such as NEOM and THE LINE are expected to be the first cities in the world built based on smart and intelligent technologies such as BIM, in all of its phases, starting from design, construction and operation, with a budget of over 500 billion USD [9].

## 2. Literature Review

### 2.1. Tender Practices and Procedure

Tender can be described as the method of winning a project by providing a service to the owner or client; this service could be tangible, such as money or assets [10,11]. Awareness of tender techniques, methods, and strategies—such as BIM strategy—may increase a contractor's chance of winning the project contract. However, not all bidding strategies are on the side of the contractor; a large number of such strategies aim to protect the owners [12]. Consequently, there is an essential need to employ a strategy that considers the project presentation and facilitates the communication between stakeholders as a primary goal [13,14]. Tender selection criteria for construction projects in Saudi Arabia depend largely on cost, with the contractor providing a minimum cost, unless it is below 70% of the owner's cost estimate. Although these notes cannot be considered advantages for the projects sought, the data are real and must be accommodated. Generally, projects in Saudi Arabia suffer from continual time delays and poor construction quality, and BIM could provide an appropriate solution to reduce the occurrence of these issues [15]. Bid prices generally are the sole basis for contractor selection around the world, not only in Saudi Arabia or the Middle East; for example, in North America and France, the lowest price bidder is selected, and abnormal prices are excluded. Conversely, in Italy, Peru, Portugal, and South Korea, the two highest-priced and two lowest-priced bids are excluded [1,16–21].

### 2.2. BIM Implementation in Tender

There are varying market responses and reactions to BIM around the globe; in the United States and Malaysia, the market is not ready for BIM, and there are worries about increasing project costs by limiting competition. Moreover, contractors are not yet well-fitted to the use of BIM, which could affect the design phase. Unlike clients, who are considered the key factor in encouraging the use of BIM, this fact can be generalized in the previously mentioned countries, despite the fact that some construction companies had used BIM effectively, especially as an important winning factor in the procurement stage. Nevertheless, there is significant progress in Canada compared to the United States in the efficient use of BIM, especially in the procurement stage [22]. However, in the United Kingdom, BIM strategy shows that it could have a direct and tangible impact in

the bidding phase. The client receives the advantages of this technology, which includes waste reduction and the avoidance of problems that could lead to huge fines on either the contractor or other projects' stakeholders [12]. This could be a strong basis for the related authorities in the United Kingdom, who have required the use of BIM strategy since 2016, to procure any project budgeted for more than GBP 5 million. Still, there are certain struggles in BIM implementation in tender process like regulation and legal considerations, which may require workshops to be developed to help contractors effectively implement BIM technology [23].

### 2.3. BIM and Project Delivery Methods

An efficient use of these technologies and methods would increase a management team's ability to make appropriate decisions in all phases of the construction process to reduce construction failures and a lack of integration and collaboration among participants in the project execution process, which could decrease the margin of risk and keep the project under budget and within costs [11]. However, previous case studies show an acceptable outcome when it comes to using Integrated Project Delivery with BIM and other risk-sharing approaches on the target price and target value of the construction industry. In fact, determinants control the feasibility of using IPD and it is concluded that certain criteria determine whether IPD or a traditional delivery method is appropriate for a particular project [24]. However, speaking within the local domain, there is no doubt that Design–Bid–Build (DBB) is the most used PDM in the construction industry, especially in the Middle East and Gulf Cooperation Council (GCC) countries. It is perceived as the simplest method of delivering projects, where all responsibilities are clearly distributed among participants and there is a minimum margin of overlap between assigned duties [25].

The American Institute of Architects leads the construction industry in the intensive use of IPD to achieve the maximum rate of return expected from the constructed facility. Moreover, the use of IPD helps architects and designers create buildings with high sustainable criteria and high energy performance, without overlooking advantages such as planning, cost estimation, cost control, and time control that IPD provides suppliers, contractors, and other project stakeholders. Additionally, IPD offers the ability to contribute to decision making in the early stages, which, for example, gives the project management team an assessment of the contractors' and suppliers' ability to complete their required activities according to the given time schedule and with the required quality; this could lead to a comprehensive time schedule [13,19,26].

Implementing BIM in any Level of Development (LOD) would be considered a BIM project. Hence, BIM implementation could be in level 300 only, in level 500, or in all levels. In practice, BIM should accommodate the selected Project Delivery Method (PDM); otherwise, BIM would not be useful if the PDM chosen by the owner cannot be fitted with the contractor's BIM strategy. Therefore, the subject of accommodating BIM to the selected PDM is discussed in this section. In the literature review, the properties and advantages of BIM and its strategy were listed to determine the best PDM to maximize these advantages [2,5,27,28]. As mentioned, several PDMs are used in construction projects, and these methods are selected based on several criteria such as the nature of the project, the number of authorities involved, the construction risks, and the client's preference. These methods largely depend on the complexity of the project; the more complicated the project, the more it trends away from traditional methods [29–31]. This occurs due to the essential need for risk sharing among project stakeholders. Methods that can distribute risk by logical doses, such as Design-Build (DB), Construction Management Multi-Prime (CM@MP), and Design–Bid–Build are considered traditional when applied to a complex project. However, IPD and CM@Risk are considered to be modified forms of DB with a different methodology for risk distribution management [24,32,33].

In general, there are a considerable amount of studies that highlight the advantages of using BIM in the construction industry: it could reduce cost, duration, and conflicts between buildings systems, and reduce waste materials and communication issues, which are all good for the client side. Nonetheless, encouraging BIM implementation would not be applicable without making it feasible for contractors to use, which requires extensive study to evaluate if BIM could be awarded to contractors in the form of project winning. Highlighting the previous issues increases the need to fill this gap, by making a comprehensive study that would evaluate BIM implementation feasibility from the contractor's perspective, by measuring increased winning probability.

### 3. Methodology

The literature review provides an understanding of the benefits of applying BIM in the construction industry, characteristics of the BIM strategy, and obstacles of implementation. Additionally, it highlights the criteria for selecting contractors during the construction bidding process. The research methodology uses a qualitative approach through face-to-face interviews as well as a quantitative approach through a structured questionnaire.

This section discusses the steps and processes required to fulfill the objectives of this paper. It includes a detailed description of the methods used to collect experts' opinions and analyze the results, and it discusses the framework design of the questionnaire to produce an outcome with a minimum margin of error. Figures 1 and 2 shows the procedure followed to satisfy the research goal, the study depends on the literature review outcomes, a survey, and interviews with experts.

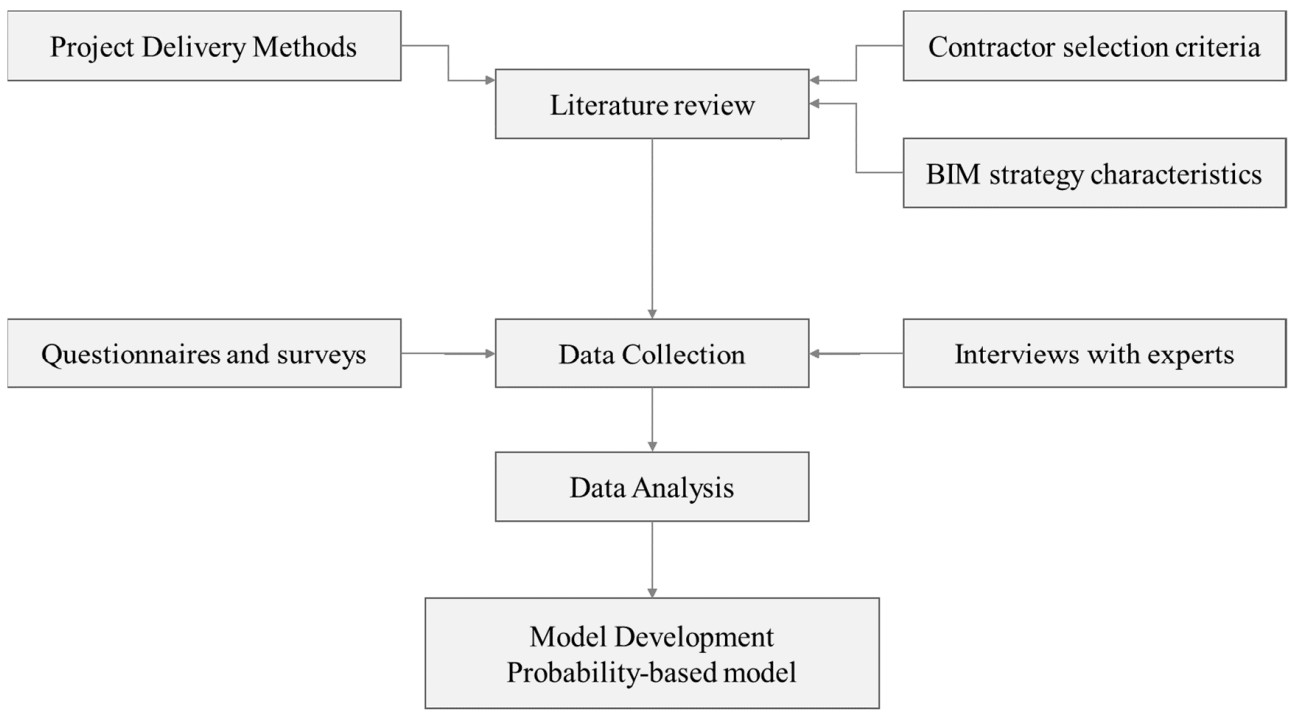

**Figure 1.** Research methodology.

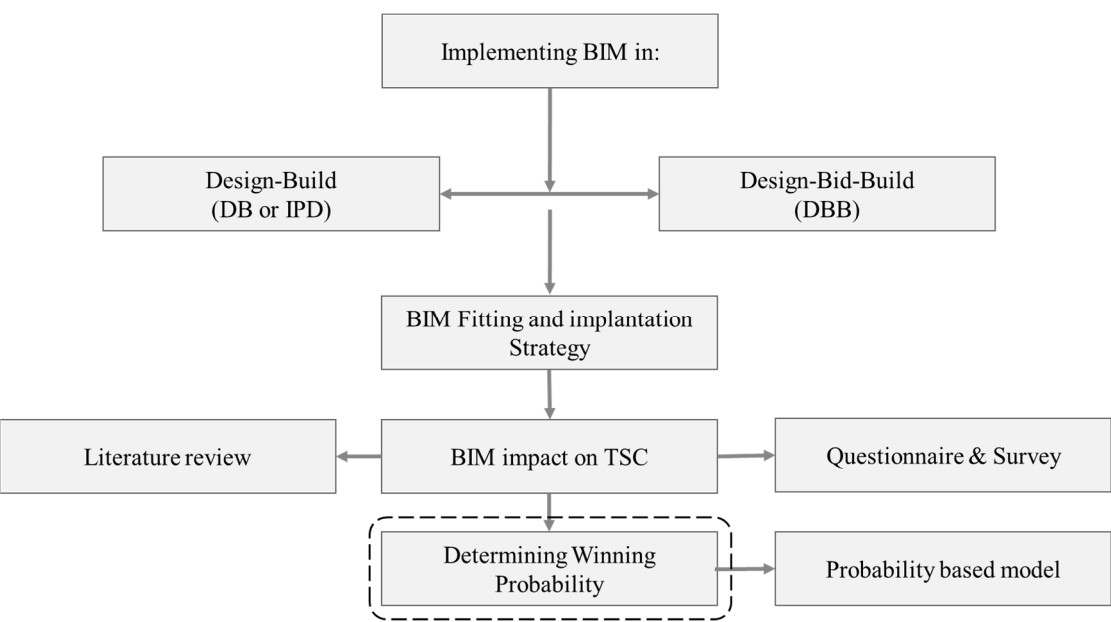

**Figure 2.** Fitting BIM with PDM.

### 3.1. Applying BIM with DBB

The latest technologies must adapt to the present conditions, and there is no need for a new technology if it is not providing tangible benefits. Therefore, a tool such as BIM is necessary to meet the current conditions and become a cornerstone for the industry progress [25]. Nonetheless, BIM as a first sight could have no significant impact on projects that use DBB and in which designers and executors are separate entities [34]. In an opposite approach, the general contractor hires technical office staff to support the site technically, transfer all Issued-For-Construction drawings to shop drawings and eventually, create the as-built drawings. Additionally, the technical office is responsible for coordinating all issues on site; in other words, a technical office is a strong tool for a general contractor to control and manage the project's subcontractors; therefore, even if the project's documents were not submitted through BIM, a general contractor who is represented by a technical office would rather transfer the documents to BIM and then extract shop drawings and produce an accurate bill of quantities, as-built drawings, and a time schedule [26,34,35]. In other words, as presented in Figure 3, the best action that general contractors can take is to use the BIM tool to their advantage in all aspects, and tendering is only one of these fields. General contractors must highlight the abilities of their technical offices to use BIM technology effectively and highlight previous projects executed successfully through BIM implementation [34].

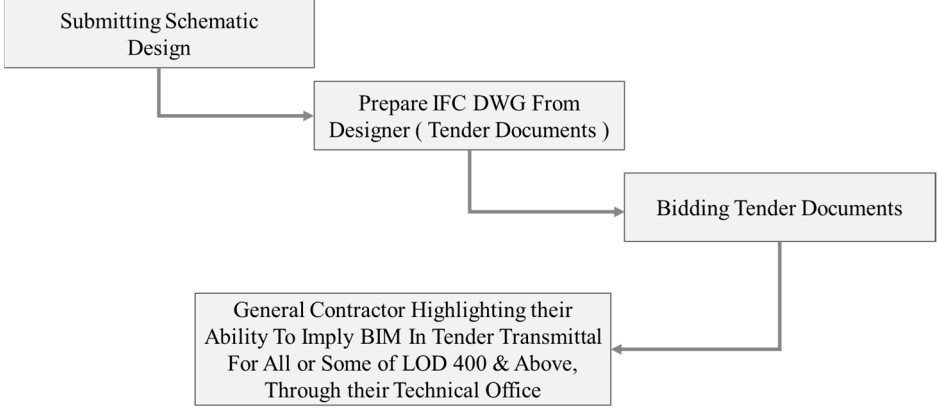

**Figure 3.** Methodology of applying BIM with DBB.

*3.2. Applying BIM with IPD*

BIM supports the management team in accomplishing the goal for which, nonetheless, BIM would be the main communication and leading tool between project participants. The key is the integration that BIM makes possible. Moreover, BIM adds value for owners in the facility management phase. When reviewing previous projects, owners might have faced information loss at the end of the construction phase. BIM can work as a method to preserve this information for facility operation and maintenance. The reviewed literature suggests that BIM and IPD can dramatically enhance project performance from conceptualization through to building management and ongoing operations. Some references refer to BIM as IPD and vice versa; they are considered to be a unified concept, unable to be separated from one [16,36].

## 4. Model Development

Based on the literature review, a questionnaire was designed to collect data. Clients and consultants are the sample for this survey, because they are the decision makers when it comes to identifying the selection criteria and assigning each criterion's relative weight. On the other hand, the feasibility of using BIM strategy is determined by contractors, who must decide whether it is feasible to apply this strategy, since it requires hiring highly trained and specialized engineers/specialists and the utilization of costly digital equipment and software packages.

First, stakeholders' opinions were collected to determine increased winning probability, by knowing their perspective on how BIM advantages could increase tender winning potential, as presented in Figures 4 and 5. However, these advantages were identified and listed according to the literature review.

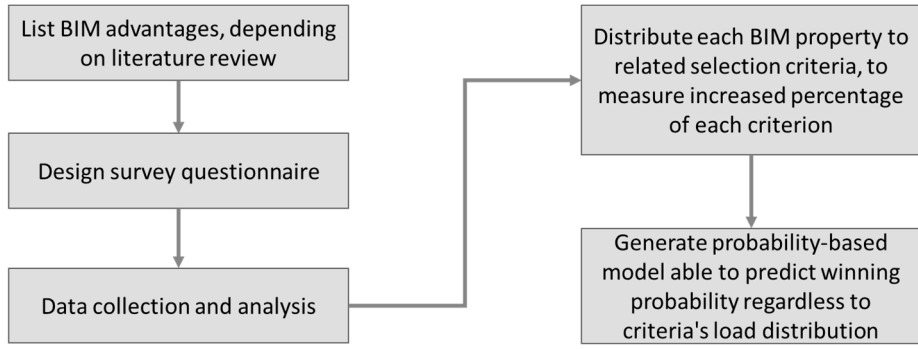

**Figure 4.** Methodology of questionnaire.

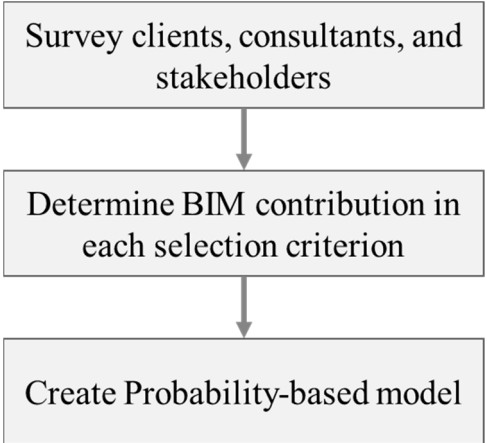

**Figure 5.** Methodology of probability-based model generation.

This survey was developed to determine the probability of winning the tender when applying BIM; it attempts to measure the impact of BIM technology on each tender selection criterion from the perspectives of clients, consultants, and stakeholders. It has a five-point scale on each side of the preferences, depending on the likelihood of choosing a contractor who applied BIM strategy with respect to those mentioned in Figure 6.

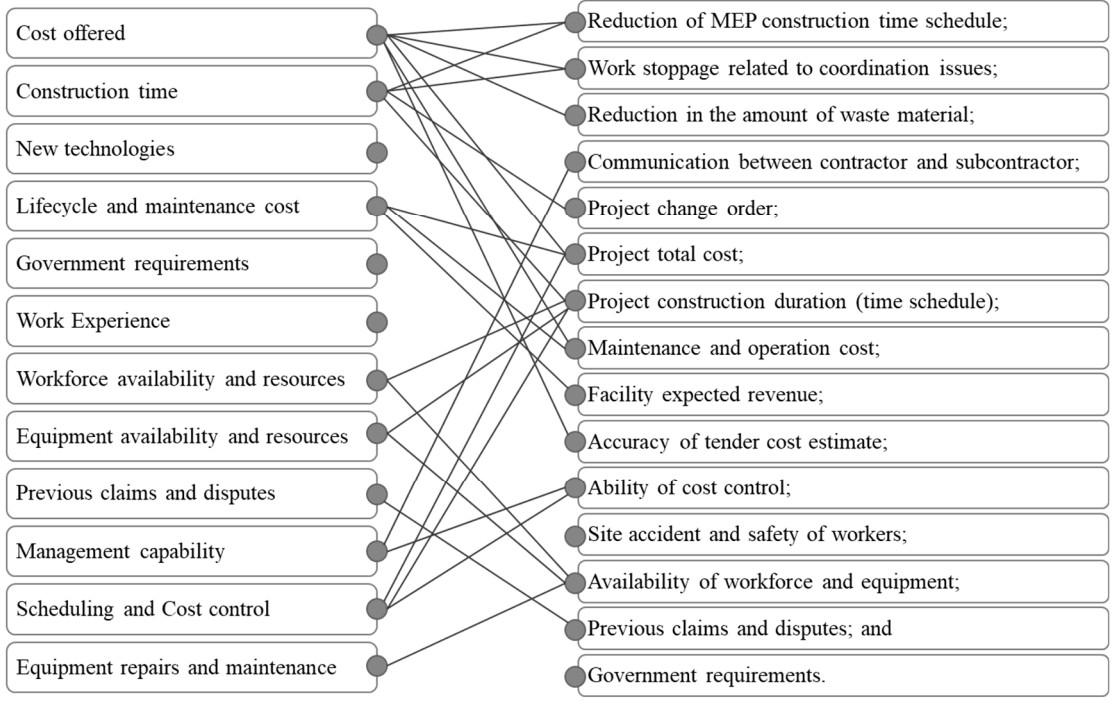

**Figure 6.** Mapping between BIM advantages and TSC.

These selection criteria were gathered from the extensive literature review, and the tendencies of preference were scaled to a likelihood degree, as indicated in Table 1.

**Table 1.** Questionnaire scaling.

| BIM Strategy | 5 | 4 | 3 | 2 | 1 | 0 | 1 | 2 | 3 | 4 | 5 | Traditional Strategy |
|---|---|---|---|---|---|---|---|---|---|---|---|---|

The questionnaire was sent to 85 engineers and specialists employed by 32 engineering firms in the Kingdom of Saudi Arabia, all of which are qualified by Saudi Aramco and the Royal Commission of Jubail and Yanbu. The sample selected for this study covered many disciplines related to building construction, such as architectural engineers, designers, quantity surveyors, BIM managers, safety engineers, civil and structural engineers, site engineers, and MEP engineers, to ensure all building disciplines were considered.

Years of experience related to building construction is a significant factor in determining the validity of this study, and consultants selected for this sample have between 7 and 35 years of experience in their specialty. Each engineer/specialist was interviewed to clarify the reason behind giving a certain grade for each criterion, to provide feedback on their opinion, to evaluate the response or to answer adequately and logically, and to guarantee no confusion was there while grading the criteria. The size of the surveyed engineering firms varied from 30 employees to 2500 employees that use BIM. All other firms that do not use this technology were excluded. Table 2 illustrates sample distribution among engineering specialties.

**Table 2.** Sample of engineers/specialists.

| Architectural Engineers | Structural Engineers | BIM Coordinators | Quantity Surveyors | Safety Engineers | MEP Engineers | Tech. Managers | Total |
|---|---|---|---|---|---|---|---|
| 31 | 16 | 8 | 12 | 8 | 10 | 5 | 85 |

## 5. Results and Discussion

The results are based on the questionnaire responses from engineers/specialists with the required expertise and include the effect of BIM on each listed criterion. The rationale behind these results was provided through interviews and the literature review. Figure 7 shows the increased percentage on each contractor selection criterion during the proposals' evaluation when contractors utilize BIM technology. It is clear that implementation of BIM positively influences the evaluation of the contractor implementing the same technology; the contractor could score more in all the surveyed selection criteria. It ranges in increased percentage from 2% to 25%. The highest increased percentage is allocated to the benefits that BIM provides to the effectiveness of the communication between contractors and subcontractors. The second highest was allocated to the ability of the contractor to conduct effective and accurate project cost control when utilizing the BIM during the construction phase.

Figure 7 illustrates the effect of BIM on each tender selection criterion. The results indicate that BIM has a strong effect on communication among project participants, maintenance and operation cost, project cost, project duration, ability of cost control, accuracy of cost estimation, and duration of MEP construction. According to the survey, BIM would increase communication with stakeholders by 25%; reduce maintenance and operation cost by 9%; reduce the project cost, time schedule for the project, and MEP construction by 8%; and reduce project change orders by 10%. Conversely, BIM seems to have a minor impact (3% or less) on previous claims, governmental requirements, facility expected revenue, and site safety.

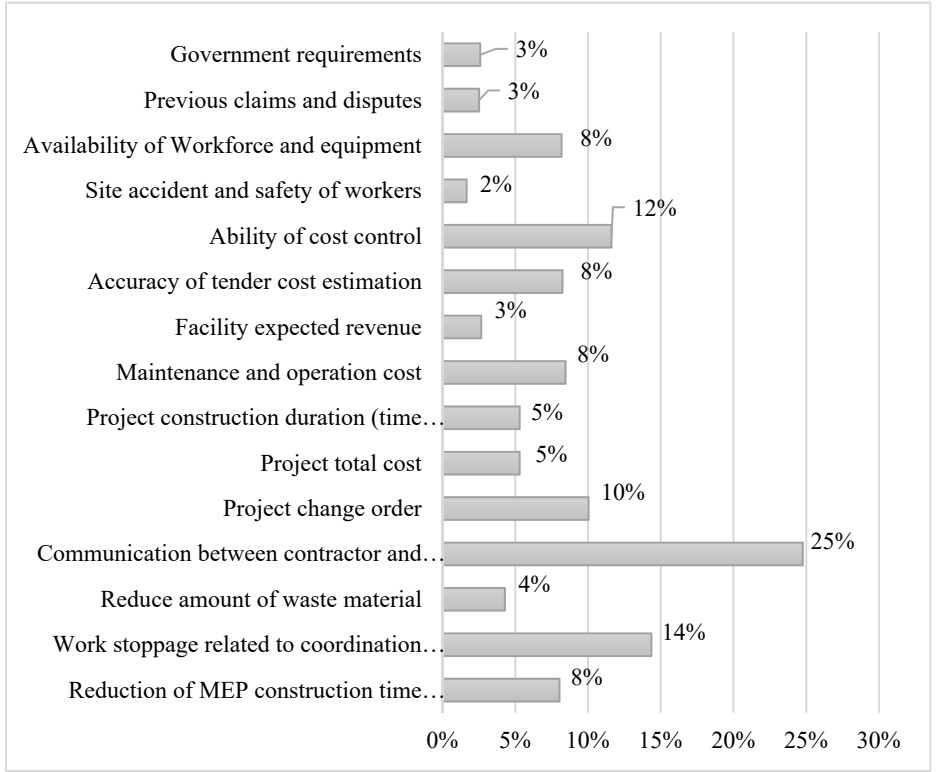

**Figure 7.** BIM impact on survey criteria.

### 5.1. Interviews Findings

(1)  Large consulting firms do not tend to use BIM, since more effort is required to transfer information to BIM software compared to smaller firms.

(2)  In mega structures, consulting engineers and designers who have little experience and background in BIM technology prefer to use traditional strategies. They use BIM in specific areas of the project where there is a misty view of systems used, and the general contractor or subcontractor can ask for more detailed drawings and specifications if the submittals do not clarify the component.

(3)  There is considerable bias for consultants who use BIM to select contractors who apply BIM technology in their tender proposal.

(4)  General contractors, represented by their technical offices and designers, prefer to use BIM in LOD 300 and above. It is not generally practical to use BIM for the schematic and conceptual design stages; in these phases, designers require fast software that renders the project roughly, to illustrate the design concept.

### 5.2. Calculating Probability of Winning

After data collection through the literature review, questionnaires, and interviews with experts, these data are analyzed to reflect the winning probability that can be obtained through AHP to generate a probability-based model that is able to predict the winning probability, regardless of the relative weights of selection criteria. After determining the experts' opinions on the impact of BIM on each criterion in the survey, the survey criteria were matched to related tender selection criteria through a logical bond extracted from the literature review. When a survey criterion had an impact on more than one contractor selection criterion, the impact margin was divided between them; then, the averages of all margins of impact from the questionnaire criteria to the selection criteria were calculated. The results represent the probability of winning the contract.

Figures 8 and 9 shows that the result of implementing BIM on the Saudi case study could increase the contract winning probability by up to 9.42% in the case of CBS.

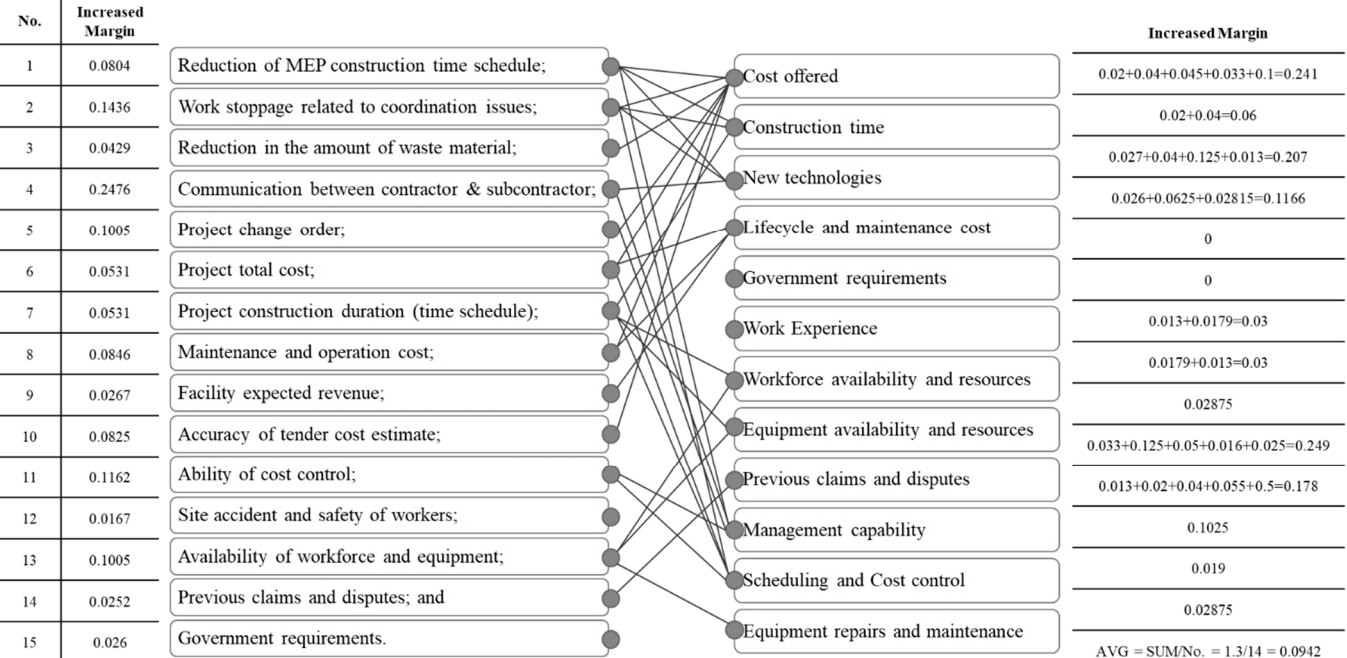

**Figure 8.** BIM impact on survey criteria calculation process.

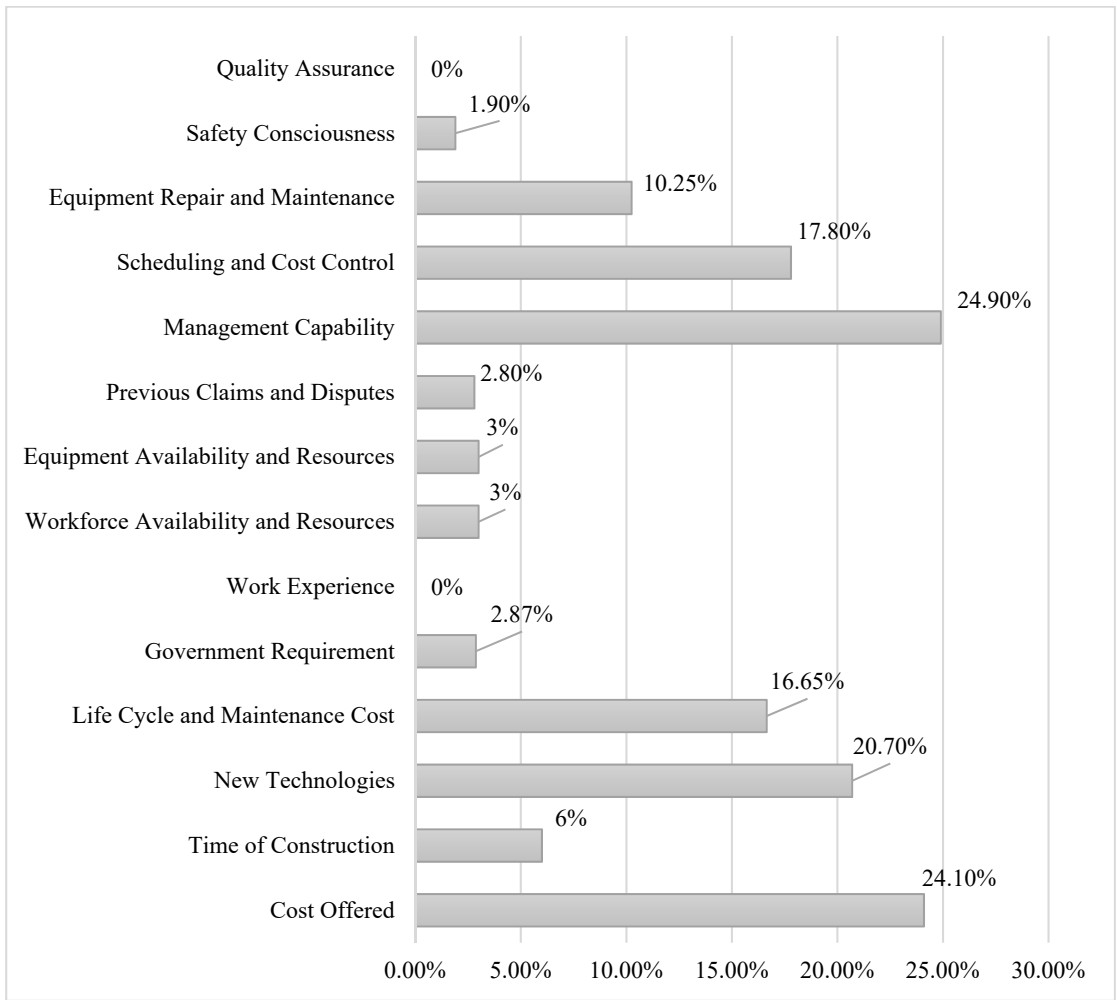

**Figure 9.** Impact of BIM on TSC.

### 5.3. Validity of Results in Practice

As was previously mentioned, there are two scenarios of biddings—QBS and CBS. In the case of QBS, previous results will be validated with minor margins of difference depending on the country; this will not be the case in CBS, in which all criteria other than cost will lose their impact. Furthermore, as mentioned in the literature review, most public sectors in many countries use cost as a sole base of selection. However, when it comes to the private sector, there are no available data that give the data of owners' tendency toward selection bases, but there is no doubt that they have extended use and awareness to select contractors on bases other than cost. Moreover, there is high votes in many countries to encourage private sector to go for QBS.

The main result of this paper is the property-based model, which is able to predict BIM winning probability regardless of the relative weight, which can be applied in any country. Nonetheless, the Saudi case study shows that the feasibility of BIM increasing the probability of winning would rely on its impact on bid price; regarding this, the core increment margin of BIM winning probability in the case of CBS is 5.5%.

Taking Saudi Arabia as a case study, it is difficult to consider a 5.5% rise in winning probability as a good way to convince contractors to use BIM, especially if the owner does not require it. Nonetheless, there should be a comparative study on the feasibility of using BIM in light of these results, taking into consideration all of the initial and running costs of transferring to BIM technology, and the risk contractors may face to learn and become professional enough to use it in all project phases. However, this percentage is applicable when proposing for the public sector that may use CBS, not in the private sector that uses

QBS, which is based on relative weight which the owner may decide. In fact, in some cases, where the priority is to finish the project on time to meet certain international events, a higher weight will be loaded to the construction duration. In other words, it depends more on bidding circumstances than where it was offered. This fact adds more validity to the probability model generated in this study; it is able to predict winning regardless of relative weight, which may vary depending on place, time, tender circumstances, or any unpredictable factor.

## 6. Conclusions

This research sheds light on the impact of BIM in the tendering process from the contractor's perspective. It aims to investigate the effect of implementing BIM strategy on the probability of winning the contract. The positive effect could encourage contractors to implement BIM in bidding, even if it is not a requirement. One of the best ways to convince contractors to use BIM is to present its ability to increase the probability of winning the tender over another contractor who does not use BIM in their proposal. However, the application of BIM in the tender could be more attractive for owners than contractors because BIM implementation guarantees high construction quality, mitigates errors and rework, and provides lifecycle cost reduction due to BIM's advantages in the operation and maintenance phases, etc.

This study uses a literature review, survey, and interviews with experts to develop a probability-based model using AHP to identify the impact of BIM on tender winning probability in various project delivery methods. The main aim of this paper is to generate a probability-based model able to predict winning probability regardless of relative weight. However, taking Saudi Arabia as a case study shows that implementing BIM strategy in the bidding proposal has a positive impact on each tender selection criterion; it could increase bid winning probability to 9.42% in the case of QBS and 5.5% in the case of CBS. Interviews with experts revealed that BIM could have a tangible impact on certain tender selection criteria, such as cost offered, facility life cycle and maintenance, management capability, scheduling, and cost control.

However, there should be a comprehensive market study to evaluate the feasibility of BIM use in light of these results, to know if it is worth going for BIM or not, especially if BIM technology is not required. Moreover, consistent studies to highlight the advantages of BIM technology for contractors is required; otherwise, extension of BIM using would be limited to initiation of owners; the best way to encourage contractors toward BIM is to highlight its financial feasibility. One way to do this is to show its impact on winning projects.

**Author Contributions:** Conceptualization, M.M. and A.E.; methodology, M.M. and A.E.; validation, M.M. and A.E.; formal analysis M.M. and A.E.; investigation, M.M. and A.E.; resources M.M. and A.E.; data curation, M.M. and A.E.; writing—original draft preparation, M.M.; writing—review and editing, A.E. All authors have read and agreed to the published version of the manuscript.

**Funding:** This research received no external funding.

**Institutional Review Board Statement:** Not applicable.

**Informed Consent Statement:** Not applicable.

**Data Availability Statement:** The data presented in this study are available.

**Conflicts of Interest:** The authors declare no conflict of interest.

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
