# Peer review of "Probability of Winning the Tender When Proposing Using BIM Strategy: A Case Study in Saudi Arabia"

_buildings, doi:10.3390/buildings11070306_

Round 1

Reviewer 1 Report

1. The paper needs to be edited to correct mistakes in English (e.g. "when proposing using BIM strategy" in the abstract).

2. As is noted in the introduction, there can be significant differences between private and public tenders. Such a differentiation should be made by the authors, as it is not clear what type of projects their study and the questions they ask deal with.

3. I don't understand what is meant by "vertebra" in Figure 6.

4. It is not clear to me who was interviewed / sent a questionnaire by the authors. They state that:

"Clients and consultants are the sample for this survey, because they are the decision makers when it comes to identifying the selection criteria and assigning each criterion’s relative weight. On the other hand, the feasibility of using BIM strategy is determined by contractors."

Indeed, the client will select the contractor, and the contractor will define the bid. Yet it seems that those interviewed / sent a questionnaire (it is not clear to me who was interviewed out of the participants in the survey) are consultants in engineering firms - who belong to neither of the previous two groups.

5. I don't understand the research question, or its methodology. A questionnaire was used to ask consultants from engineering firms to rank advantages of BIM, which were selected based on those reported in literature. Those apparent advantages were then linked with equivalent criteria for selecting contractors for tenders. However, there are many different ways in which BIM can be implemented in projects, and other ways in which this can be presented in bids. How a client selects a contractor in a tender is yet another question, which depends on the tendering method and criteria, as well as on the contractor's proposal. In other words - this proposal, including any reference to the future use of BIM in the project, is not the same as what will actually be done in the project, and the advantages BIM will provide. All of this is not discussed or clarified, which leaves the reader with too many questions. 

6. Obviously, none of the findings are validated, something that I would consider a minimum requirement.

Reviewer 2 Report

Article summary

The reviewed paper submitted to the Buildings journal tries to deal with winning bidding strategies for construction works in light of the potential use of BIM in projects, however not as much as it sounds. In fact, this manuscript looks promising but it is not ready for publication in the presented form. Its contribution is not explicit. A body of the manuscript discusses only KSA circumstances which is not bad but it needs more explanation why this case is so unique and how to repeat the success strategies in other countries?

Main impressions (Novelty? Interesting? Sufficient impact? Adds to the knowledge base?)

In general, the article is incomplete, certain elements are confusing for a reader (e.g. what does a final probability score - 9.42% - mean for the tenderers? what if they all implemented BIM? etc.) and make it impossible to recommend it for publication. an article in a prestigious journal. I would recommend the Authors address a revised text with a list of recommendations for key market players or to rebuild it, make it closer to construction management matters and resubmit it again elsewhere.

Specific comments and suggestions

A title is quite adequate but sounds artificial. Please, avoid abbreviations.

An abstract is quite unclear, does not present any motivations to publish the text in Buildings. Includes many repetitions (“selection criteria” line 21, 22) or errors (kingdom of Saudi Arabia - lowercase).

An introduction is not consistent, seems to be a little bit clumsy.

Methods are not well described. How literature review was performed? Are the Authors sure they could not create other drawings showing clearly the differences between DBB and DB (figures 4, 5)? Are the Authors sure Figure 7 should include “create a probability-based model” two times? Results are not clear and sometimes repeated twice (table 3 and figure 9, figure 11 and table 4), conclusions are too short, and therefore seem incomplete too.

Language should be revised.

References are OK.

All in all, the manuscript focuses only on KSA standards, without any attempt to generalize the problem, so for this reason, it cannot be published in this shape.

Final recommendation:

  • RECONSIDER AFTER MAJOR REVISION

Reviewer 3 Report

The research aims to “study the impact of using Building Information Modeling 14 (BIM) in the tendering process from the contractor’s perspective”. It is a good topic. Actually, it is well known that the advantages of using BIM benefit to construction, which would help contactors efficiently deliver projects. On the other hand, it looks like this research investigates the acceptable probabilities of clients to these benefits. The followings are the main comments:

  1. Please clearly define the differences between using BIM and BIM strategy.
  2. Section 2 Literature review is not well analysed. Please find the points such as the research necessaries and research gaps by summarising and discussing previous studies. One paragraph discusses one publication is not a good way to do this.
  3. More contents in 5.1 and 5.2 shall be provided and discussed, but I think the contents in Section 3 and 4 could be reduced. The key issue is to deeply discuss your findings.
  4. For the research methodology, why is the literature review used? If it is used for questionnaire design, please build the literature connections between them.
  5. Poor format; e.g. Table 3. Please check the reference styles.
  6. Please rewrite conclusions, such as research contributions, limitations, and future works.

Round 2

Reviewer 2 Report

The improvements are fine.

Reviewer 3 Report

Satisfied with the paper.